# A Comparison of the Effects of Ultrasonic Cavitation on the Surfaces of 45 and 40Kh Steels

**Dmitriy S. Fatyukhin \***[ID]**, Ravil I. Nigmetzyanov, Vyacheslav M. Prikhodko, Aleksandr V. Sukhov and Sergey K. Sundukov**[ID]

Department of "Technology of Construction Materials", Moscow Automobile and Road Construction State Technical University (MADI), Leningradsky Prospect, 64, 125319 Moscow, Russia; r.nigmetzyanov@madi.ru (R.I.N.); v.prikhodko@madi.ru (V.M.P.); a.sukhov@madi.ru (A.V.S.); s.sundukov@madi.ru (S.K.S.)
\* Correspondence: d.fatyuhin@madi.ru

**Abstract:** The ultrasonic treatment of metal products in liquid is used mainly to remove various kinds of contaminants from surfaces. The effects of ultrasound not only separate and remove contaminants, they also significantly impact the physical–mechanical and geometric properties of the surfaces of products if there is enough time for treatment. The aim of this study was to compare the dynamics of ultrasonic cavitation effects on the surface properties of 45 (ASTM M1044; DIN C45; GB 45) and 40Kh (AISI 5140; DIN 41Cr4; GB 40Cr) structural steels. During the study, changes in the structure, roughness, sub-roughness, and microhardness values of these materials were observed. The results showed significant changes in the considered characteristics. It was found that the process of cavitation erosion involves at least 3 stages. In the first stage, the geometric properties of the surface slightly change with the accumulation of internal stresses and an increase in microhardness. The second stage is characterized by structure refinement, increased roughness and sub-microroughness, and the development of surface erosion. In the third stage, when a certain limiting state is reached, there are no noticeable changes in the surface properties. The lengths of these stages and the quantitative characteristics of erosion for the considered materials differ significantly. It was found that the time required to reach the limiting state was longer for carbon steel than for alloy steel. The results can be used to improve the cleaning process, as well as to form the required surface properties of structural steels.

**Keywords:** ultrasound; cavitation; erosion; roughness; sub-roughness; structure; microhardness

## 1. Introduction

Acoustic cavitation, based on the effects arising after imparting high-frequency vibrations into a processed medium, is an effective way of influencing the surface layers of metal products.

Acoustic cavitation consists of the formation of bubbles caused by the rupture of the continuity of the liquid under the action of variable sound pressure. When a spherical cavitation bubble collapses, shock waves arise, accompanied by instantaneous pressures and temperatures, the calculated values of which can be up to several hundred MPa (1) and several thousand degrees (2) [1–3].

$$P_{max} = P_{min} \left( \frac{R_{max}}{R_{min}} \right)^{3\gamma} \tag{1}$$

$$T_{max} = T_{min} \left( \frac{R_{max}}{R_{min}} \right)^{3(\gamma-1)} \tag{2}$$

Here, $R_{max}$ is the maximum bubble radius at the initial moment of collapse, $R_{min}$ is the minimum bubble radius at the moment of the end of the collapse, $P_{min}$ and $T_{min}$ are the

pressure and temperature, respectively, at the initial moment of the collapse, and $\gamma$ is the indicator that determines the state of the gas in the cavity (1–1.33).

In addition to shock waves, cumulative jets arising from the collapse of the cavitation bubble near the surface have a significant effect on the treatment process. In such cases, the spherical shape of the bubble becomes unstable and its different parts move at different speeds. The most distant part of the bubble surface from the treated surface moves with maximal speed. The study of this process using high-speed shooting [4] made it possible to establish the speeds of the bubble side in the direction of pushing from the hydrodynamic jets, reaching 500–600 m/s. This process also creates pressures up to hundreds of MPa.

Recent studies have shown the possibility of forming non-equilibrium plasma inside the cavitation bubble. In such cases, the electron temperature depends mainly on the vibration frequency, reaching 8000 K at 20 kHz and 12,000 K at 1057 kHz [5].

The considered effects cause plastic deformation and erosion of the treated surface. In such cases, the effect of the action of one bubble is microscopically small, although repeated cyclic action leads to significant results, i.e., cavitation erosion is of a fatigue nature. This is why the processing time is one of the main criteria for optimization of the ultrasonic treatment process.

When using frequencies over 345 kHz, the effect of cavitation is minimized and the treatment takes on an erosion-free character (without the formation of damage to the treated surface). In such cases, the main mechanisms of action on the surface are microflows formed by pulsating bubbles [6].

Additionally, one of the factors influencing the efficiency of ultrasonic liquid treatment is the presence of various inclusions in the liquid. For example, one study [7] showed that cavitation bubbles are better generated in heterogeneous systems. At a frequency of 20 kHz, solid particles contribute to the activation process.

In general, the course of all ultrasonic technological processes based on cavitation erosion treatment can be described on the basis of a kinetic curve linking the loss of specimen mass with the treatment time [8–10]. Therefore, the treatment process can be divided into three stages:

- The first stage is an incubation period, whereby the cavitation energy is spent on plastic deformation, structural transformations, and other changes in the surface layer that do not cause mass loss. This period characterizes the resistance of the material to cavitation;
- The second stage involves a cavitation erosion treatment, leading to geometric surface changes due to the formation of erosion pits and accompanied by an increase in mass loss;
- The third stage involves the destruction of the surface when brought to the critical state. This is expressed as a significant area of erosion damage, which practically does not change during further treatment.

The durations of the stages and the magnitude of the effects are determined by the parameters of the ultrasonic treatment mode, the initial properties, and the structure of the treated material.

The technology involved in ultrasonic liquid treatment has been actively developed since the 1950–1960s [11–15], and has seen wide use in fields ranging from engineering to medicine.

Currently, the process is well studied and a significant number of publications have been devoted to it, including articles on the physics and mechanics of the process [11,14–16], the application of treatment at high vibration amplitudes [16,17], the choice of technological medium [16,18,19], physical and mathematical modeling [20–22], and the features of the calculation and selection of emitting instruments [23], among others (a small range of publications is given as an example).

At the same time, the overwhelming majority of studies on the topic have been devoted to ultrasonic cleaning, which is necessary to carry out in the intervals of the incubation period so as not to cause damage to the surface. Although cleaning is the most studied

(the most significant studies were published before the 2000s) and widely used ultrasonic technology, a number of modern studies on this topic can be found.

For example, research results regarding the development of ultrasonic cavitation in cold water were presented in [24–26]. The results showed that the maximum cavitation activity was observed in the range of 7–20 °C, which was explained by a decrease in the vapor pressure contained in the liquid. In [27], it was proposed to intensify the cleaning process by introducing microbubbles of air—obtained using the method of hydrodynamic cavitation—into the liquid. The authors of [28] stated that the main disadvantage of ultrasonic cleaning is the need to immerse the product to be cleaned in a container with a process liquid. To solve this, a cleaning method was proposed, which consists of the use of cavitation in a sheet of a detergent solution applied to the surface. In [29], an installation was proposed based on the movement of a rod oscillating system along the cleaned surfaces of large and geometrically complex products. The technology used to increase the area of the product to be cleaned was also considered in [30]. It was shown that in addition to traditional cleaning mechanisms (collapsing bubbles and acoustic streams), it is possible to use the bubble formation energy, whereby bubbles are coalesced into multibubble clusters. To activate the energy stored in the clusters, an additional ultrasonic action on the technological medium with a low-amplitude vibration source was proposed. Another study [31] aimed to improve understanding and optimization of the cleaning process. This article compared the effects of cleaning with gas and vapor bubbles separately. The observations showed that vapor and gas bubbles were effective in cleaning high- and low-adhesion contaminants, respectively. The authors of [32] carried out experimental studies on the removal of a support material obtained by 3D printing products from materials that were UV-cured. The use of ultrasound made it possible to treat hard-to-reach areas of the locations of the supports. Another study [33] was devoted to the application of ultrasonic liquid treatments to clean ceramic membranes.

Issues related to the influence of cavitation–abrasive treatment on the properties of the surface layer have been and continue to be investigated, mainly relying on the cavitation resistance of the materials, which function under field conditions that contribute to erosive destruction of the surface (propellers, parts of ships, pumps, hydraulic equipment, etc.). Contemporary articles have been devoted to the development and study of new cavitation-resistant materials (including non-metallic materials), coatings, methods of material production using additive technologies, and ultrasonic treatment of the melt [34–43].

An analysis of these studies, as well as those performed earlier [7–10,44–48], showed that the cavitation resistance of a material is determined by many factors, which include the chemical composition and microstructure of the material, the tendency for strain hardening, the microhardness of the structural elements, and the initial geometry of the surface.

Far fewer articles have been devoted to the study of the influence of cavitation as a way to ensure the required surface characteristics. This direction has been actively developing in recent years in accordance with modern trends in technology development, which ties in with the reduction of the sizes of products, the use of complex geometry, the use of multifunctional coatings, the use of products in extreme conditions, and the constant improvement of existing technological processes. In this regard, the scope of the studies has increased to cover both the effects of ultrasonic liquid treatment on surface properties at the micro- and nanolevels and the effects of these properties on the operation of specific types of products.

The ultrasonic treatment of aluminum alloy 6061-T6 (GOST 4784-97: AD33; EN 573-3: EN AW-6061), when pretreated with a laser [49], leads to the removal of fiber-like structures and a decrease in the roughness parameter Ra by about 10%.

A previous study [50] on the effects of ultrasound on the surfaces of metal plates with a cubic-centered (Al, Ag, Cu) and hexagonal (Zn) structure showed increases in the roughness parameter (Rtm), whereby after treatment of Al for 4 min the Rtm was over 30 μm, after treatment of Cu for 5 min the Rtm was 17.9 μm, and after treatment of Ag for 5 min the Rtm was 13.3 μm.

In [51], the process involved in changing the surface of a cobalt alloy after the implantation of nitrogen ions with HIPed Stellite 6 was studied in detail. The wear mechanism is based on plastic deformation of the cobalt matrix, which starts from the interfaces between the $Cr_7C_3$ and Co-matrix. The avulsion of carbides creates cavitation pits, which initiate the propagation of cracks along the cobalt matrix, ending with the avulsion of a massive piece of material. As a result, the roughness parameters Sa and Sz increase.

The production of surface nanofoams from titanium dioxide by ultrasonic treatment at a frequency of 20 kHz was considered in [52]. Ultrasonic treatment led to four morphological stages: untreated titanium, a mesoporous layer of titanium dioxide, nanoribbons, and a hierarchical structure. The sub-microroughness values of the specimens (in the same order) were 9.79, 12, 4.39, and 7.67 nm. The instability of the sub-microroughness values during treatment was explained by the relationship between the mechanical destruction of the surface and the temperature effect.

In [53,54], the effect of ultrasound treatment on the etching process of silicon in an isopropyl alcohol solution was considered. As a result, the surface roughness of Si (111) plates decreased from Rq < 5 nm to Rq = 1 nm, with increases in alcohol concentration and ultrasound power.

The efficiency of the use of ultrasound for etching has been proven in the treatment of PCV plastics [55]. The results in [55] showed an increase in the Cu–PVC adhesion by 13% compared to etching with chromic sulfuric acid. This was connected with the formation of pits with a radius range of 0.03–20 microns, which improved the conditions for the attachment of the copper coating. Meanwhile, the roughness increased by up to 1.5–2-fold.

The issues involved in reducing the surface roughness of metal products obtained using additive technologies and ultrasonic treatment were considered in [56,57]. The fact that the working medium contained cavitation bubbles made it possible to treat any surface type.

In [58], the effect of the cavitation treatment on the surface roughness was associated with the ratio of the bubble size to microroughness. The following cases were considered: the bubble was larger than the microroughness; the sizes were commensurate; the microroughness was greater than the bubble. A sharp increase in roughness was noted when the sizes were commensurate. In this case, the cavitation type was slot cavitation, which is characterized by sharp increases in erosion in small gaps.

Studies on the effect of cavitation on 45 steel (DIN, EN:C45; GB:45; ASTM:M 1044) were presented in [59]. Long-term ultrasonic treatment led to grain refinement to a depth of 50 microns and the creation of a hardened layer of the same depth.

Ultrasonic treatment is used for surface modification in the medical industry. For example, in [60], the influence of ultrasound as a method for repairing the surfaces of metal titanium implants was studied. After treatment with ultrasound at a frequency of 25 kHz for 10 min, the topography of the specimens changed, with a decrease in the size of rough and wetting angles.

The inverse approach was presented in [61]. The effect of the roughness of the emitter surface on the dynamics of cavitation bubbles was considered, rather than the effect of ultrasound on the roughness. During the testing of a smooth surface, many bubbles were formed, which grouped into a cloud (cluster). The number of bubbles obtained when testing the rough surface was much lower, as they behaved erratically and accumulated into small clusters or vibrated alone.

Thus, the use of ultrasonic liquid treatment leads to changes in the properties and geometry of the surface at the micro- and sub-microlevels. In such cases, depending on the initial conditions and parameters of the processing mode, both an increase and decrease in roughness are possible, which can be used for various technological purposes. Cavitation erosion can be considered not only from a negative point of view as a mechanism of surface destruction, but also as a mechanism for the formation of surfaces with certain properties. For example, in [55], erosion pits increased the surface area, which increased the adhesion between the base and the coating.

In this work, studies on the use of ultrasonic cavitation erosion treatment for the widely used 40Kh (DIN, EN: 41 Cr 4; GB: 40 Cr; AISI: 5140) and 45 (DIN, EN: C45; GB: 45; ASTM: M 1044) structural steels are carried out in order to study the possibility of changing their surface properties.

## 2. Materials and Methods

### 2.1. Materials

To carry out the experimental studies, cylindrical specimens measuring 10 mm in diameter and 5 mm thick were cut from rods of 40Kh and 45 structural steels. After cutting, specimens were normalized at T = 860 °C. The chemical compositions of the steels are shown in Table 1.

**Table 1.** Chemical composition of steels 40Kh and 45 (%).

| Material | C | Cr | Mn | Ni | Cu | W | Si | Fe |
|---|---|---|---|---|---|---|---|---|
| 40Kh | 0.39 | 0.95 | 0.57 | 0.26 | 0.09 | 0.01 | 0.2 | 97.53 |
| 45 | 0.46 | 0.09 | 0.55 | 0.27 | 0.1 | 0.01 | 0.21 | 98.31 |

Two specimens of each material were fixed in an aluminum mandrel and poured with protacryl. After the hardening of protacryl, polished sections were prepared. When specimens were removed, one of them remained as the control and the second was subjected to ultrasonic treatment.

### 2.2. Experiment Scheme and Equipment

The ultrasonic liquid treatment of polished sections was carried out according to the scheme shown in Figure 1.

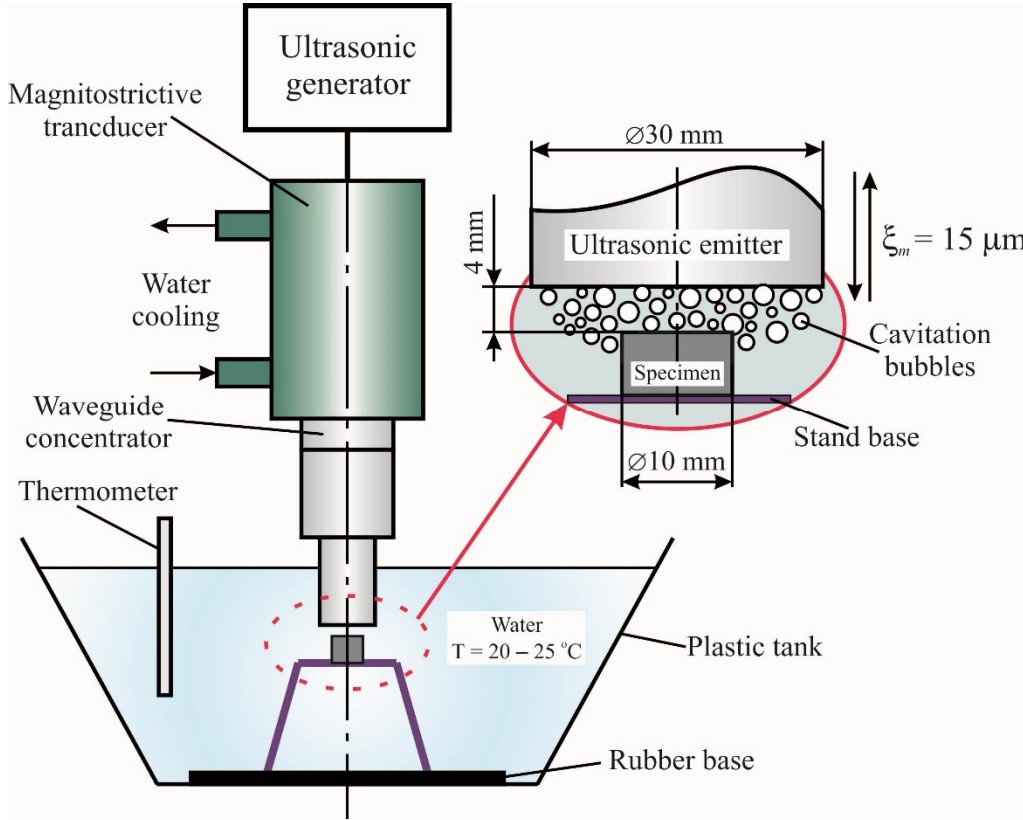

**Figure 1.** Ultrasonic treatment scheme.

For processing, a PMS-2.0-22 rod three-half-wave magnetostrictive oscillatory system (LLC "Inlab", St. Petersburg, Russia) was used. This consisted of a magnetostrictive transducer made of 49Fe49Co2V alloy, located in a water cooling jacket, with a titanium alloy waveguide concentrator soldered to its end. A stepped titanium emitter with an emitting surface diameter if 30 mm, with a vibration amplitude increase coefficient $k_y = 4$, was connected to the waveguide by means of a threaded connection.

The oscillating system was powered by a UZG2-22 generator (LLC Apfalina, Moscow, Russia), with a maximum output power of up to 2 kW, which is required to achieve high oscillation amplitudes. The generator had frequency and amplitude automatic control functions. This is important for carrying out long-term treatments, since the heating of the emitter and temperature increases in the treated medium lead to changes in the resonant frequency of oscillations.

The emitter was immersed in water until a distance of 4 mm to the specimen was reached. The treatment was carried out at a vibration amplitude m = 15 μm, which was set before emitter was sunk in water by calibrating it with a dial indicator. The resonant vibration frequency was f = 21.5 kHz.

The acoustic and technological parameters of the ultrasonic treatment mode were selected due to the following considerations.

The low-amplitude treatment was carried out using vibration amplitudes m < 10–12 μm (for water). This process is characterized by the transition of random sections of the volume that is treated by ultrasound into cavitation and the almost absence of large-scale acoustic streams.

When the amplitude exceeds 10–12 microns (high-amplitude mode), a sharp development of the cavitation area occurs at the end surface of the emitter. At this time, a strong absorption of acoustic energy occurs. As a result, directional hydrodynamic streams are formed. The formed streams lead to the formation of a stable cavitation area, which can be divided into three zones:

- The first zone coincides with the cavitation area, which joins directly to the surface of the emitter and geometrically fits into a cylinder with a diameter corresponding to the diameter of the emitter and a height of 3–10 mm;
- The second zone is behind the first and has a truncated cone shape, the height of which is determined by the depth of penetration of the acoustic stream (up to 100 mm);
- The third zone contains the rest of the treated medium, which is outside the first and second zones and shows practically no erosion activity.

In the first zone, only the cavitation processing mechanism is involved. This is characterized by high erosion activity. At the same time, the distribution of the vibrational speed and pressure are relatively homogeneous.

The second zone involves a mixed mechanism of action. There is a significantly smaller number of cavitation bubbles, which are carried by the acoustic stream in the direction away from the emitting surface, and the erosion activity is reduced.

In this way, with the accepted experiment scheme, the specimen surface is located in the zone of intense cavitation, which has high stability and predictability during the processing. To prevent the specimen from moving during ultrasonic treatment, it was glued to a stand, which was installed on a rubber base.

During the experiment, with long periods of treatment (over 10 min), the water temperature was maintained in the range of 20–25 °C to maintain constant cavitation.

Based on preliminary experiments, the treated specimens were analyzed after 10, 20, 30, 40, and 60 min of treatment and then every 60 min. The treatment finished when the roughness parameter Ra changed by less than 5% over three time intervals in a row.

### 2.3. Estimation of Structure and Surface Properties

The parameters of the control and treated specimens that were estimated were the micro- and sub-microstructure, roughness and sub-microroughness, and microhardness values.

The microstructure was researched on a METAM PB-22 metallographic microscope (LOMO JSC, St. Petersburg, Russia), which is an inverted microscope with a desk on top. This microscope is designed for the visual observation of the microstructures of metals, alloys, and other opaque objects in reflected light under direct illumination in bright and dark fields. The magnification range is 80×–1000×. Roughness parameters were measured on a Model 130 profilometer (Proton JSC, Zelenograd, Russia).

The operation of the profilometer analyzing the roughness of the measured surface with an inductive sensor at a constant speed. The sensor probe is a diamond needle. Then, the movement of the probe is converted from an analog to a digital signal with further processing of the signal on the computer. The tracing length is up to 12.5 mm, while the limits of the measurement accuracy of the roughness parameter are Ra = ±(0.002 + 0.03 Ra).

Certain roughness parameters given in this study correspond to EN ISO 4287:1997, including the arithmetic mean deviation (Ra), ten-point height (Rz), and maximum peak-to-valley height (Rtm), while the others correspond to GOST 2789-73, including the average pitch of the profile roughness (Sm), the average pitch of local peaks (S), and the relative reference length of the profile (tp, where p is the level of the profile section). Additionally, using a profilometer, the oil capacity parameter Q was determined. This parameter is not a roughness parameter, although it is entirely determined by the surface microrelief and is a characteristic of the actual area of the researched surface. Therefore, the oil capacity is important during coating, as it will determine the adhesion strength of the coatings.

The sub-microstructure of the surfaces was analyzed using an SMM-2000 scanning probe microscope (Proton JSC, Zelenograd, Russia) with atomic force microscopy. This approach is used to measure the geometric and physical parameters of the surfaces of specimens with nanometer dimensional resolution and without vacuum conditions.

For this study, an MSCT cantilever was used, which had 6 beams of various lengths and stiffnesses. The height of the needles on the beams was 3 μm, while the radius range of the tip was 300–600 Angstroms.

The images of the surface topography were obtained during the scanning process by controlling the parameters and the scanning process.

The sub-microroughness values were assessed through processing and analysis of the images obtained from the microscope.

The microhardness values of the specimens were measured on a PMT-3 microhardness tester (JSC LOMO, St. Petersburg, Russia), which operates by indenting a Vickers diamond tip with a square base and a tetrahedral pyramid into the test material. This provides geometric and mechanical similarity to indentations as the indenter deepens under load.

Numerical and graphic processing of the measurement results was carried out in Statistica software. While the dependences of the roughness parameters were plotted, the values were taken as the arithmetic means of five measurements in cavitation-treated areas.

## 3. Results and Discussion

### 3.1. Microstructure Changes

Metallographic photos of the surface of specimens, which characterize the dynamics of changes in the microstructure under the action of cavitation erosion treatment, are presented in Figure 2 for 40Kh steel and in Figure 3 for 45 steel.

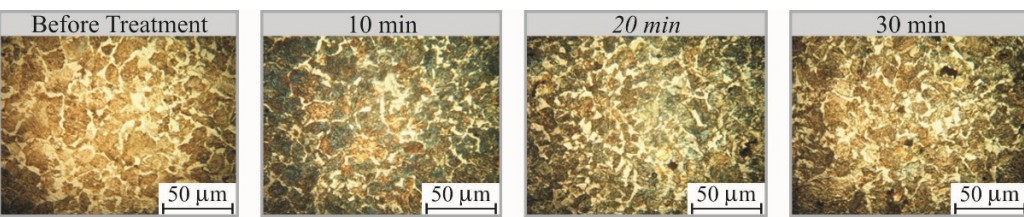

**Figure 2.** *Cont.*

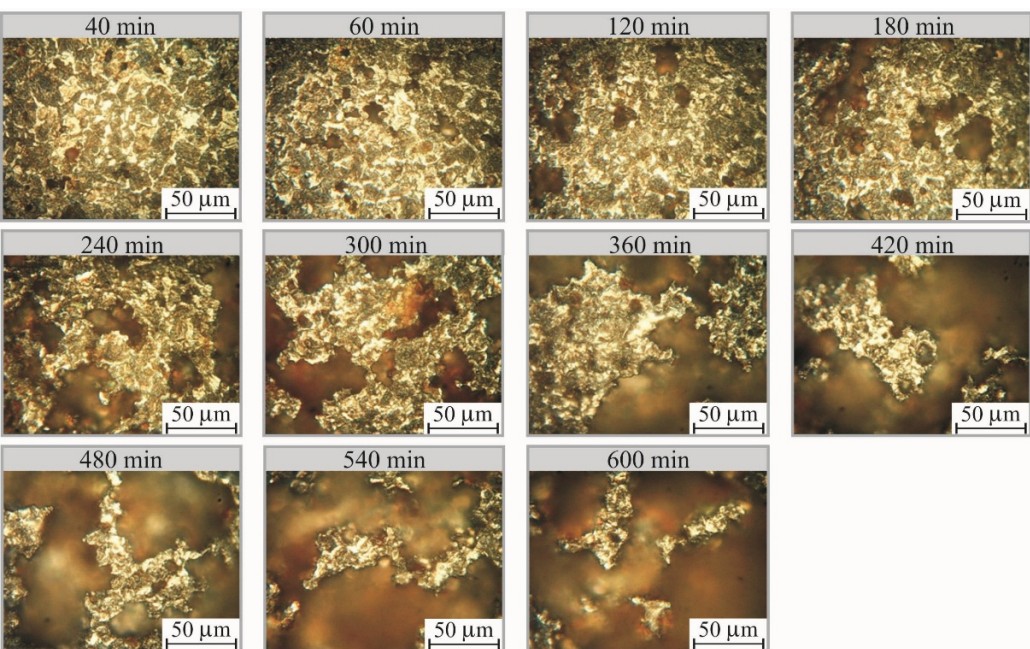

**Figure 2.** Dynamics of changes in the microstructure of 40Kh steel after ultrasonic treatment.

Before treatment, it can be seen that 40Kh steel had a typical ferrite–pearlitic structure typical of hypoeutectoid steel. During the first 10 min of cavitation treatment, no changes occurred. The first signs of erosional damage appeared after 20 min of treatment. With further treatment up to 60 min, an increase in the number of erosion pits occurred, which formed at the grain boundaries.

This can be explained by peculiarities of the distribution of the cavitation activity over the treated surface. Usually, the places of greatest impact are microrelief inhomogeneities, i.e., areas where more bubbles accumulate in the largest peaks and valleys. Since a smooth polished surface was treated here, the most convenient place with inhomogeneous properties was the grain boundary. As a result, most of cavitation bubbles collapsed along the grain boundaries, providing a shock effect that led to deformation and destruction.

After the formation of the first pits at the grain boundaries, these changed into inhomogeneities, which was conducive to high-cavitation activity. Further action in these places led to the erosional destruction of grains. This became noticeable after 60 min of treatment—the size and shape of some pits matched the shape of the grains. Therefore, upon further treatment for more than 60 min, the main mechanism for the growth of the area of erosional damage was an increase in the size of the previously formed pits. However, in this case, the formation of new pits also occurred.

The increased erosion damage seen in the period from 60 to 540 min of treatment was similar to an avalanche and the damage occupied more than 80% of the specimen area by 540 min. At this moment, the surface reached a certain limit state and further treatment did not lead to changes in surface roughness.

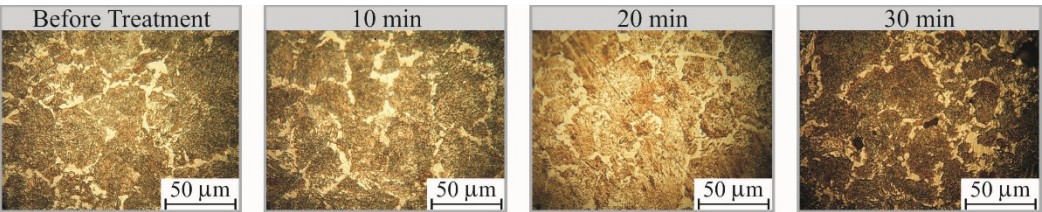

**Figure 3.** *Cont.*

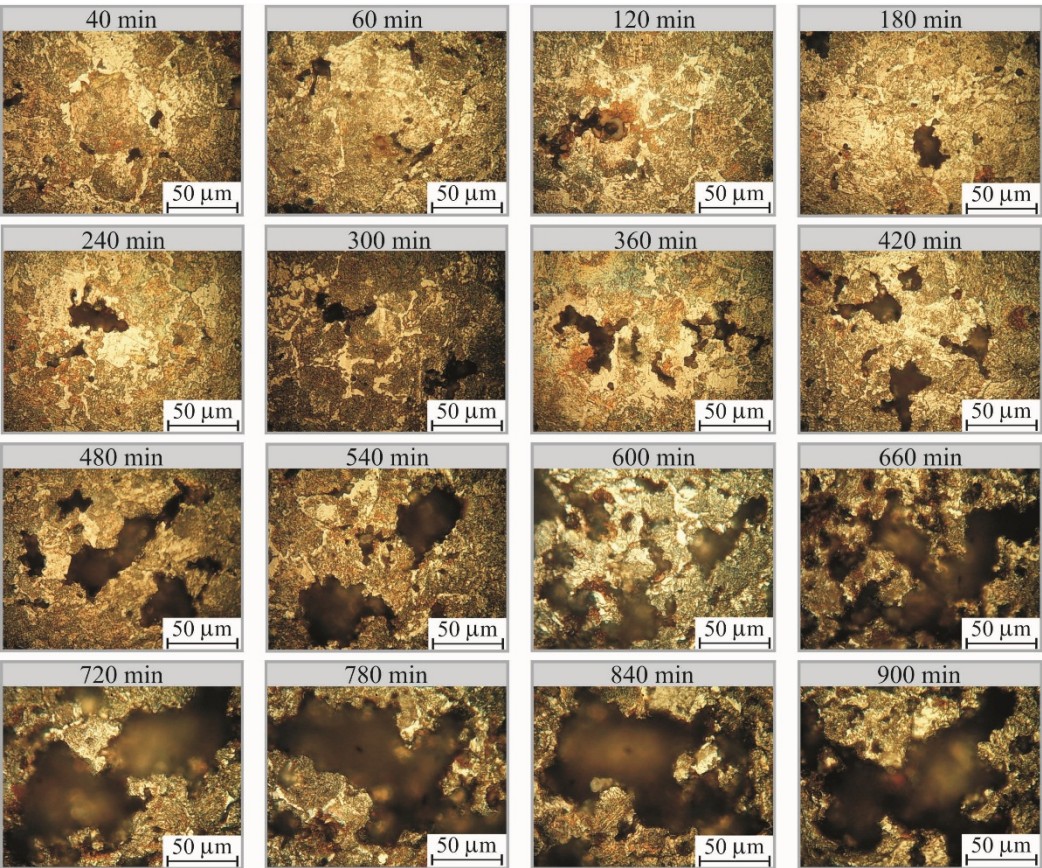

**Figure 3.** Dynamics of changes in the microstructure of 45 steel after ultrasonic treatment.

It can be seen that the 45 steel before treatment had a similar structure to 40kH steel, with a slight increase in the proportion of pearlite. However, the growth in erosional damage showed some differences. The first erosional pits arose by 20 min of treatment, when the structure was not refined. Further treatment led to an increase in the number of erosion pits, which were also located at the grain boundaries. The first traces of the erosional destruction of the grains appeared only after 180 min of treatment. During the subsequent ultrasonic treatment for up to 420 min, the erosional destruction of the grains increased, although there was not an abrupt increase in the sized of the formed damage. This growth, as well as the merging of erosion foci, began at 480 min and ended at 900 min of treatment, when the surface reached the limiting state. At this moment, the area of damage reached 60% of the area of the specimen.

Thus, for the two considered steels of the same class with similar properties, the changes in the microstructure under the action of cavitation treatment differed in nature. Firstly, it can be seen by the presence of the 45 steel (180–480 min) that despite the beginning of the erosional destruction of grains, there was no significant increase in the area of erosional damage.

This fact can be explained by the different tendencies of the materials under consideration regarding strain hardening. Figure 4 shows the microhardness distribution profiles from the surface to the depth of the material for specimens treated in the limited state.

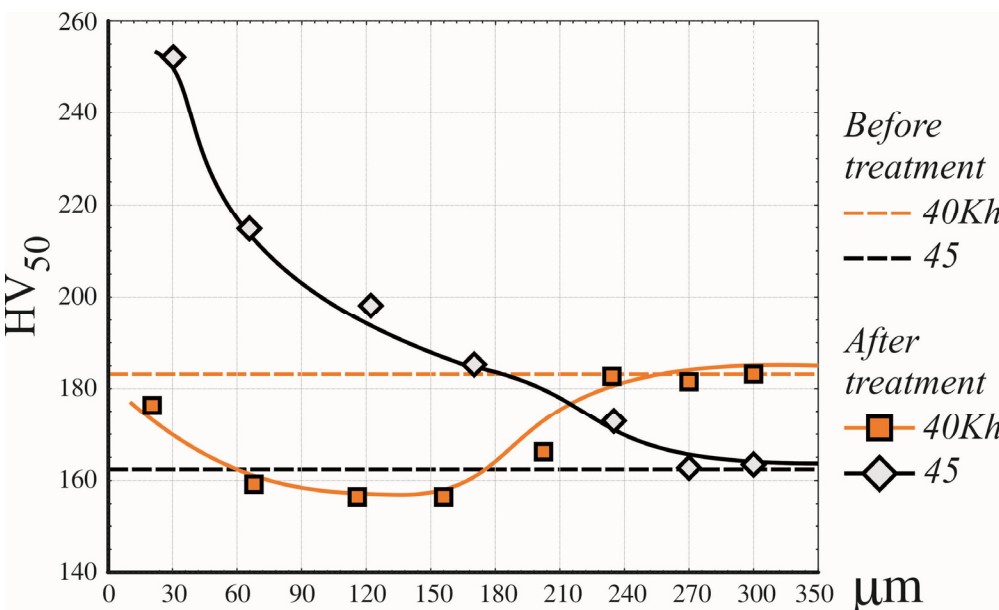

**Figure 4.** Microhardness profiles of specimens of 45 and 40Kh Steels.

As a result of cavitation, the 40Kh steel specimen did not show a pronounced hardened layer, although a decrease in microhardness was noted at a depth of 150 microns, which was a consequence of the erosional destruction of the grains. In 45 steel, in contrast with 40Kh steel, a hardened layer was formed at a depth of 150 microns, where the microhardness values exceeded the initial values by 1.5 times. Then, the microhardness decreased evenly to values equal to the control values at a depth of 250 microns.

The obtained data indicate that 45 steel, unlike 40Kh, is subjected to hardening during cavitation treatment, and the energy is spent on surface deformation during the treatment period from 180 to 480 min. When 40Kh steel is treated, energy in the same period is spent on further destruction of the surface and an increase in the area of erosion pits.

These statements were confirmed during the analysis of the results on the change in roughness.

### 3.2. Influence of Ultrasonic Treatment on Roughness Parameters

Erosion damage caused by cavitation leads to significant changes in the surface roughness.

Figures 5 and 6 respectively show for 40Kh and 45 steels the dynamics of the changes in the roughness parameter Ra under the action of cavitation erosion treatment, the evolution of the roughness profiles at different points of treatment, as well as the corresponding black-and-white images of the microstructures. The images were taken at a lower magnification compared to the previous section to allow better visualization of the growth of the erosion area on the surface. Photos of the specimens before and after treatment are also presented.

The analysis of the Ra (t) for the 40kH steel relation shows that the changes in roughness occur in 3 stages. In the first stage (0–240 min), there is an insignificant increase in the Ra parameter from 0.04 to 0.74 μm. The second stage (240–540 min) is characterized by an abrupt increase in the rate of roughness growth, whereby Ra increases almost 7 times to 5.2 μm after 300 min of treatment. In the third stage (from 540 min), the roughness does not change.

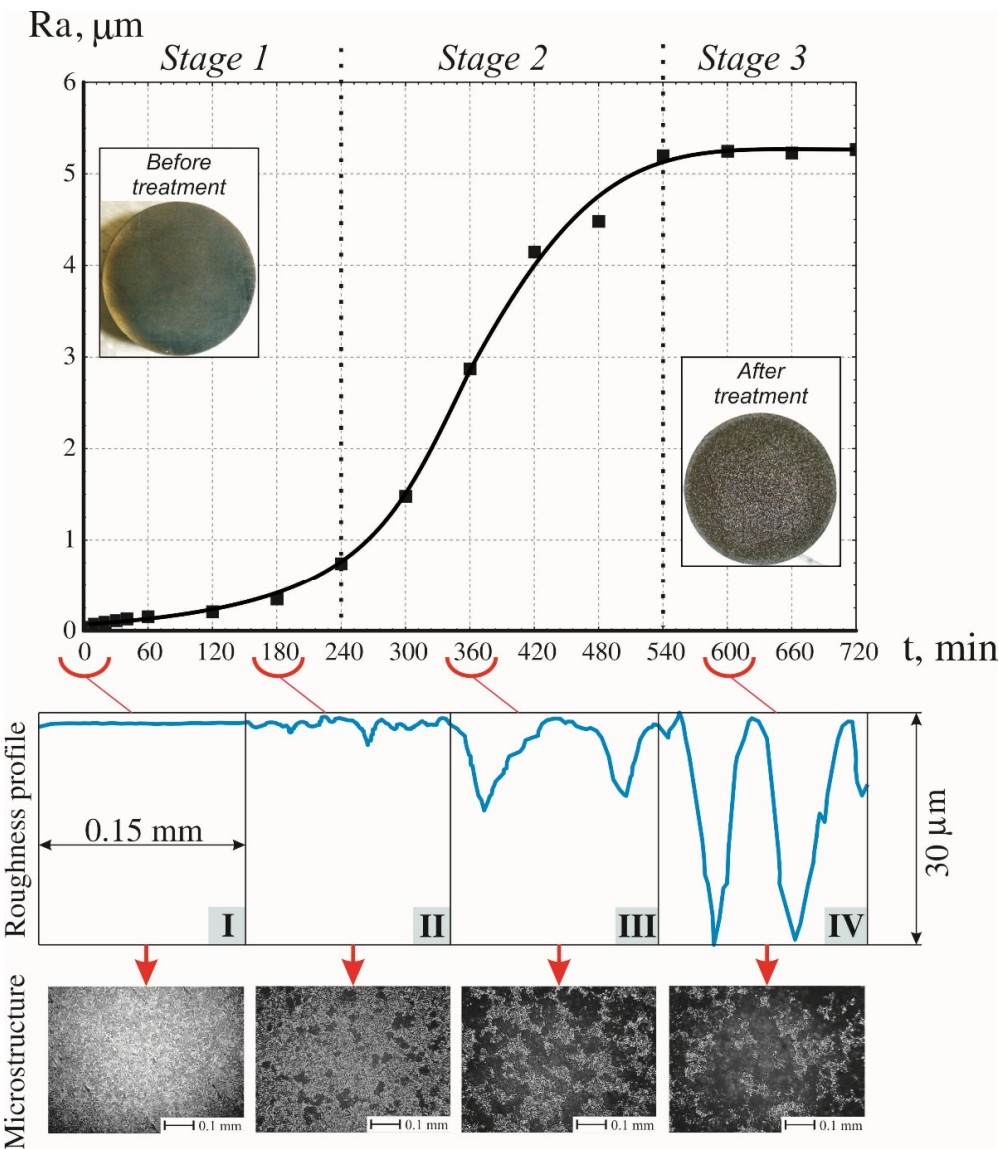

**Figure 5.** Dynamics of changes in roughness, Ra, for 40Kh steel.

Comparing the dynamics of the Ra parameter with the progress of surface erosion, a number of features can be identified:

- The growth in roughness occurs during processing before the start of the erosion. This can be explained by the fact that at the initial moments of treatment under the action of cavitation, oxide sheets are removed (as considered in [62]), the surface is cleaned from residual grinding products (as considered in [63]), and individual surface elements are deformed (as considered in [64]);

- A significant increase in roughness can be observed starting from 240 min of treatment, while an active increase in the area of erosion damage occurs starting from 60 min of treatment. This is due to the start of the erosion processes deep inside the specimen, i.e., when cavitation acts inside an already formed large cavity (up to an area of several grains). As a result, damage of the next lower layer of grains begins to occur. The abrupt rise in the graph is probably related to the size of the cavity area, which creates optimal conditions for the occurrence of "slot" cavitation, the erosion activity of which is extremely high;

- The stabilization of the Ra (t) graph after 540 min of treatment coincides with the end of the growth of the erosion area.

The influence of the considered factors can be traced while considering the changes in roughness profile with the course of the treatment process. Before treatment, the specimen was a polished section with a smooth surface (I). During treatment, erosion pits began to form and individual erosional destruction of the grains occurred, leading to the formation of roughness, which was characterized by peaks and valleys alternating in small steps (II). Further, the damage grew and merged with other erosion hotspots, the cavitation activity spread deep into the specimen, and the depth and width of valleys of roughness increased significantly (III). When the growth of the erosion area became slower, it spread further into the specimen body until reaching a certain moment. Then, the roughness was characterized by the maximum depth of the valleys without showing increased width (IV).

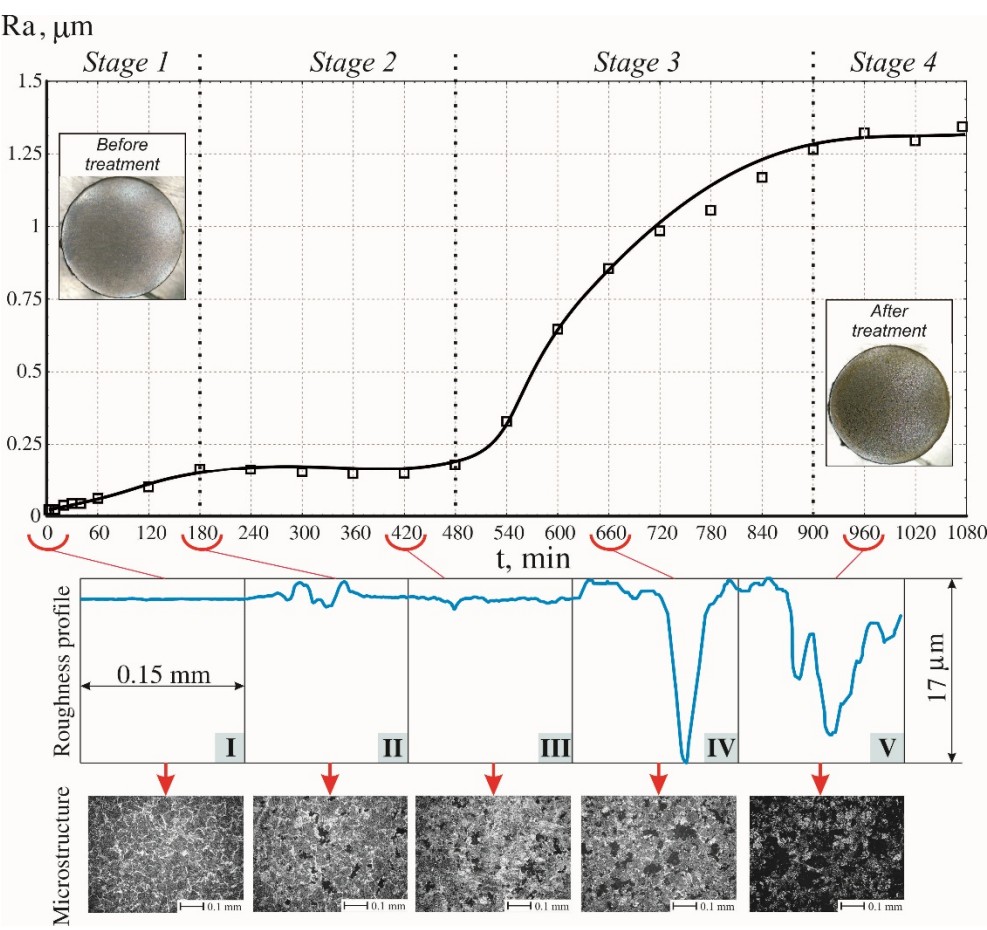

**Figure 6.** Dynamics of changes in the roughness parameter, Ra, for 45 steel.

The Ra (t) function for 45 steel differs in that the changes in roughness occur in four stages. Compared to Ra (t) for 40Kh steel, 45 steel has a stage (from 180 to 480 min of treatment) when the roughness almost does not change and not even a slight decrease can be traced (stage II). This stage coincides in time with the period of strain hardening of the surface described above, which explains the slight decrease in the Ra parameter. This also explains the lower values of Ra for 45 steel. The remaining stages of the formation of roughness are identical in nature to 40Kh steel and are similar in time.

Differences between the steels under consideration can also be traced to the dynamics of changes in the roughness profiles. For 45 steel the following stages occur: (I) polished section; (II) the beginning of the erosional destruction of grains; (III) deformation and hardening of the surface, leading to a slight decrease in roughness; (IV) abrupt increases in the area and depth of erosion pits; (V) the limiting state of the surface, where the profile is characterized by the largest width of the valleys and the absence of erosion penetration deep into the specimen.

The dynamics of changes in other measured roughness parameters (Rz and Rtm, Figure 7) for both steels are largely correlated with the Ra parameter. The differences are the greater relative increase in the values of parameters at the initial stage. This is due to the fact that these parameters are calculated on the highest peaks and valleys, where the effect of ultrasonic cavitation is more pronounced.

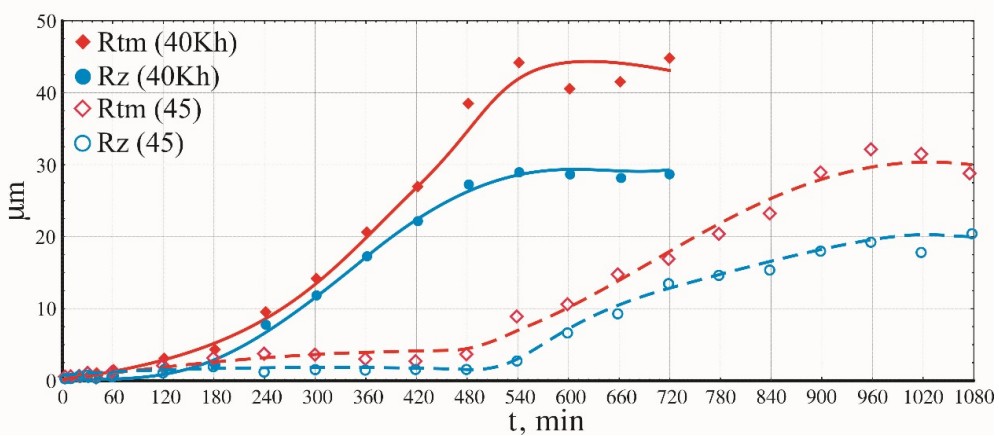

**Figure 7.** Dynamics of changes in the roughness parameters Rtm and Rz.

The functions of step parameters of roughness Sm(t) and S(t) (Figure 8) can also be divided into three stages. An abrupt increase in Sm coincides with the active start of the erosional destruction of the grains, the formation of erosion accumulations, and their further growth. A significantly smaller change in the parameter S is caused by the fact that the parameter S is calculated without reference to the center line of the roughness profile, which is significantly reduced in the treatment relative to the initial line of peaks. As a result, global surface changes associated with erosion and the erosional destruction of grains do not significantly affect its values.

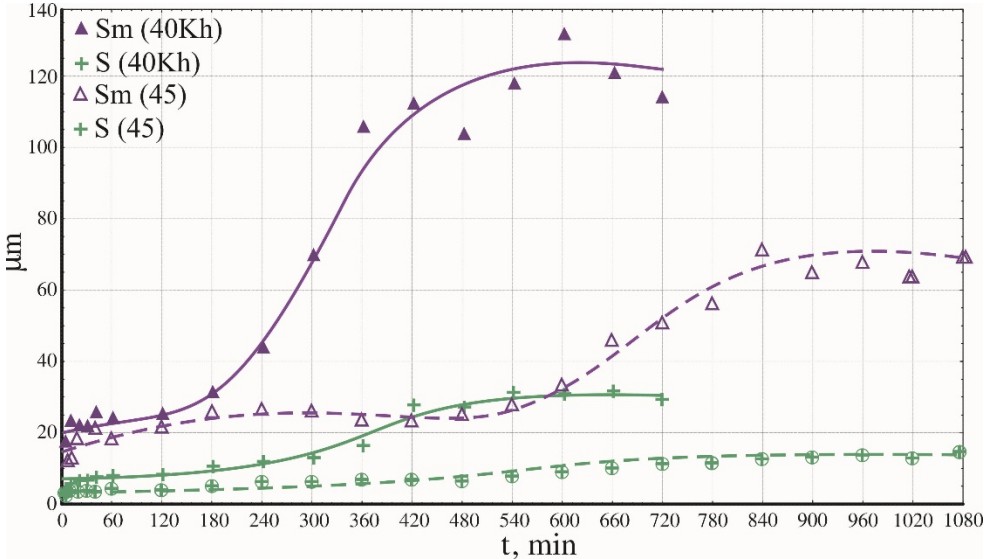

**Figure 8.** Dynamics of changes in the roughness parameters Sm and S.

The values for the relative reference length of the profile presented in Figure 9 are calculated here to provide the level of the profile section $p = 66\%$. The values of the parameter predictably decrease in proportion to the increase in the area of erosional damages and spread into the depth of the specimen.

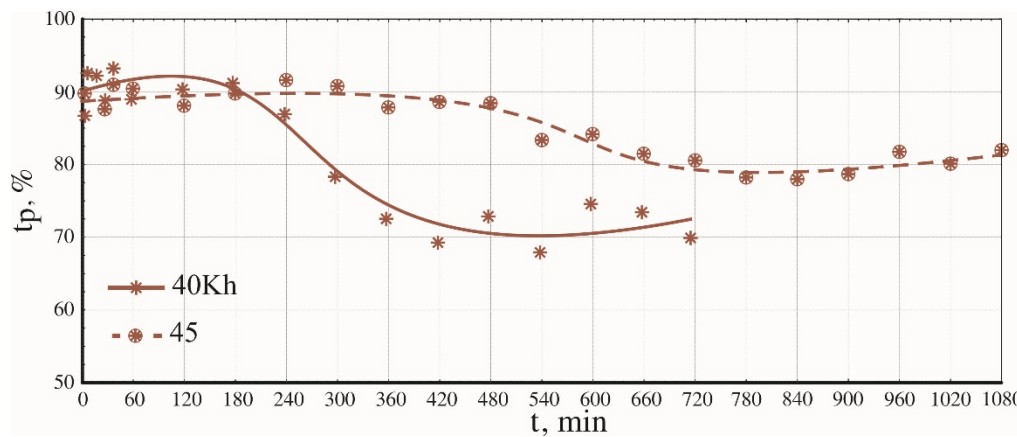

**Figure 9.** Dynamics of changes in the roughness parameter tp.

Figure 10 shows the dependence of the change in oil capacity Q on time during cavitation treatment.

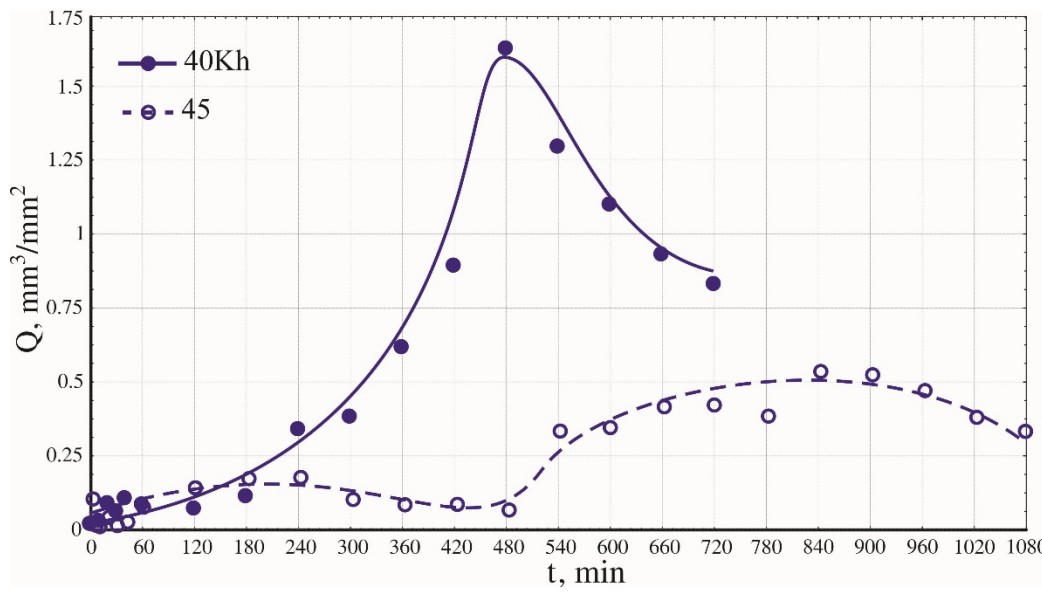

**Figure 10.** Dynamics of changes in surface oil capacity.

A noticeable extremum can be traced using the function Q (t), which was observed during processing for 480 min for 40Kh steel and for 540–600 min for 45 steel. These times correspond to the moments when erosion does not lead to changes in the roughness parameters. Microdamages and surface deformations occurring at the beginning contribute to the increased oil capacity. Subsequently, when large particles of a material begin to separate from the surface, the actual surface area decreases, which leads to decreased in oil capacity.

### 3.3. Study of Sub-Microstructure

The 3D images of the surfaces were obtained using the constant height method, which involves maintaining the fixed end of the cantilever using the microscope scanner at a constant height [13]. The obtained images measuring 308 × 308 nm are presented in Figures 11 and 12.

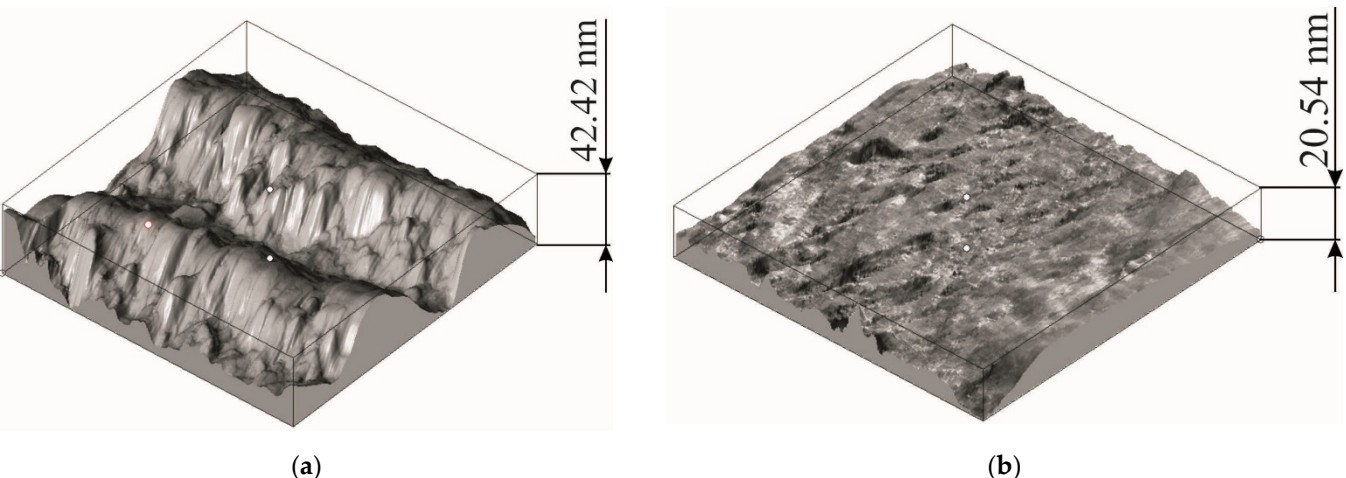

**Figure 11.** Surface topology (atomic force microscopy) of specimens of 40Kh steel: (**a**) before treatment; (**b**) after cavitation treatment.

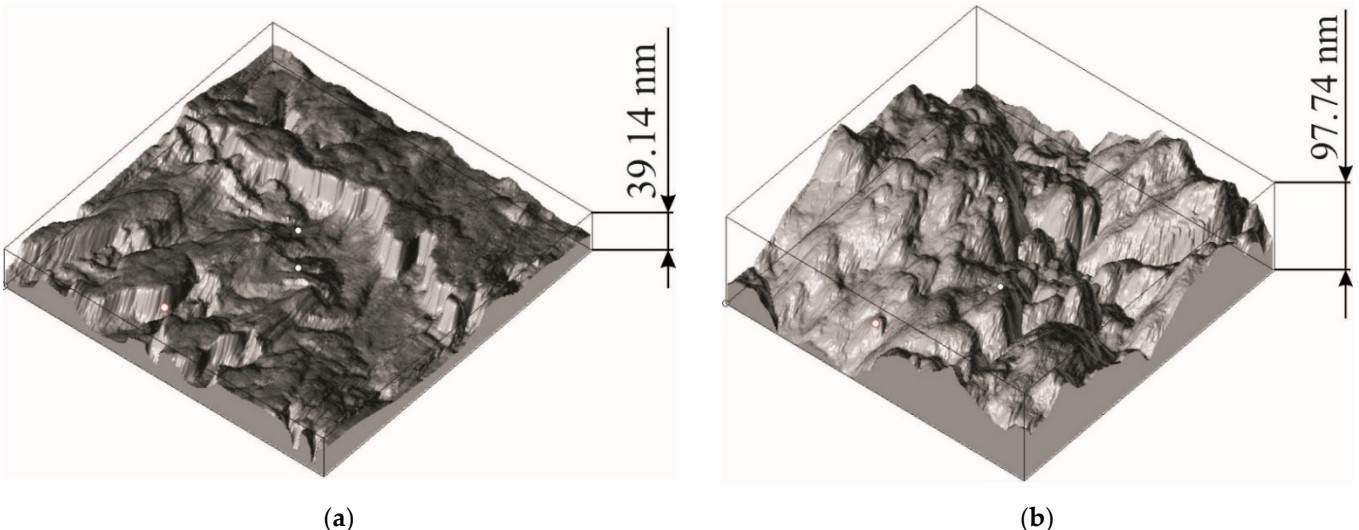

**Figure 12.** Surface topology (atomic force microscopy) of specimens of 45 steel: (**a**) before treatment; (**b**) after cavitation treatment.

Analysis of the obtained results showed opposite reactions for the surfaces of 40Kh steel and 45 steel at the sub-micro layer. Before ultrasonic treatment, the sub-microstructures of the steels were characteristic of the grinding process. After treatment, the surface of the 40Kh steel was burnished, causing a decrease in the height of roughness by 5–6 times, while the surface of 45 steel, on the contrary, showed a rougher relief with an increase in height parameters by about 3 times.

Apparently, the obtained differences were also caused by the different natures of the occurrence and growth of erosional damages. The smooth surface of 40Kh steel is a consequence of the erosional destruction of grain particles or the erosional destruction of whole grains, whereas the valleys on the surface of 45 steel are caused by deformation from cumulative jets and shock waves. These differences were also confirmed by the nature of the roughness in Figure 11b, where the valley on the left side of the surface shows a deformed area, while the roughness spreading from it to the right contains an influx of metal displaced from the deformation zone.

## 4. Conclusions

During the course of the experimental studies, the influence of ultrasonic cavitation erosion treatment on the surface dynamics of 40Kh and 45 structural steels was studied. The obtained results led to the following conclusions:

1. The growth of cavitation erosion along the treated surfaces for the materials under consideration showed significant differences, which were mainly pronounced during the hardening period in 45 steel. The main factors influencing the research results were the ability of chromium to segregate impurities to grain boundaries, the large proportion of ferrite in the 40Kh steel, and the reduction in impact strength with the addition of chromium. As a result, 40Kh steel showed a "weak" boundary inclined to accelerated destruction;

2. A comparative analysis of the dynamics of the development of erosional damages in the steels and the resulting roughness profiles made it possible to consider in more detail the development of cavitation erosion and to clarify the stages of initiation and evolution of erosion damage. For 40Kh steel, the treatment can be divided into 3 stages, while for 45 steel it can be divided into 4 stages;

3. The dynamics of the changes in the roughness profile for 40Kh steel were established, whereby the area of damage first increases, then the depth increases. For 45 steel, on the contrary, the maximum depth is first reached, then the area increases;

4. With an increase in the area of erosional damage, the step parameters of roughness mostly increase, while with an increase in the depth of damage, the height parameters increase;

5. From the point of view of ensuring the surface roughness of the materials under consideration, the most effective stage is the abrupt increase in the height of roughness. Thus, it is possible to change the Ra parameter from 0.04 microns to 5.2 microns for 40Kh steel, and from 0.02 microns to 1.3 microns for 45 steel;

6. These results can be used in the treatment of complex-shaped products to obtain a specific roughness, as well as in the development of ultrasonic cleaning processes.

**Author Contributions:** Conceptualization, D.S.F. and V.M.P.; methodology, S.K.S. and A.V.S.; software, R.I.N.; validation, D.S.F. and S.K.S.; formal analysis, R.I.N. and A.V.S.; investigation, S.K.S. and D.S.F.; resources, R.I.N. and A.V.S.; data curation, S.K.S. and D.S.F.; writing—original draft preparation, S.K.S.; writing—review and editing, D.S.F.; visualization, A.V.S. and R.I.N.; supervision, V.M.P.; project administration, V.M.P.; funding acquisition, V.M.P. All authors have read and agreed to the published version of the manuscript.

**Funding:** This research was funded by the Russian Science Foundation, grant number No. 21-19-00660.

**Institutional Review Board Statement:** Not applicable.

**Informed Consent Statement:** Not applicable.

**Data Availability Statement:** The data presented in this study are available on request from the corresponding author.

**Conflicts of Interest:** The authors declare no conflict of interest. The funders had no role in the design of the study; in the collection, analyses, or interpretation of data; in the writing of the manuscript, or in the decision to publish the results.

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
