# Peer review of "A Comparison of the Effects of Ultrasonic Cavitation on the Surfaces of 45 and 40Kh Steels"

_metals, doi:10.3390/met12010138_

Round 1

Reviewer 1 Report

Paper describes an interesting issue regarding the surface treatment using ultrasonics. Two popular steel samples were investigated in relation to their surface development under cavitation loads. Moreover, the effect of cavitation on the hardness of the surface layer was also shown. The paper balanced on the edge of surface treatment and cavitation erosion, in turn, presents interesting and worth publishing results regarding the metallic material behaviour under cavitation loads. It can be seen that authors put a lot of effort to conduct this experiment and the result develops the state of art in the field of cavitation erosion process characterisation.

Paper scope suits Metals and presents interesting results regarding the ultrasonic cleaning of steel surfaces degradation, also partly describing the erosive behaviour. On the other hand, I believe that the authors should improve the quality of this paper and will correctly address my suggestions.

My comments on the paper:

Major comments

  1. I am worried about the sub-roughness measurements. In my opinion, it should be deleted because it gives no important information.  I am almost sure that the analysed area was too small to show the surface topography. Also the presented results contradict the fact, that surfaces were polished before testing. This type of steel, especially after annealing has a relatively coarse grain size. Thus the area of investigations should be wider than each of the grain sizes. Please see the following work regarding AFM usage for roughness description of fine-grained alloy https://doi.org/10.3390/ma14092324 In your case the AFM can be employed if the measurement area will be enlarged. Now, results are affected by a relatively high error of measurement. See that in table 1 you obtained Ra exceeding 11um which is impossible for polished surfaces. Please improve this
  2. Provide metallographic evidence for grain refinement or improve this phrase "
    The first signs of erosional damage appear after 20 min of treatment, while some
    refinement of the structure is also noticeable.
    304

    "
  3. Please compare the microstructures of the investigated grades. 40kH steel should contain some carbides. How do differences in chem composition influence ultrasonic results? What are the differences in the Rm or Re of annealed steels and how do they influence the changes of the roughness? Comment on that.
  4. This phrase regarding the kinetics of the cavitation erosion process must be explained: "
    at the initial moments of treatment under the action of 389
    cavitation oxide sheets are removed, the surface is cleaned from residual grinding 390
    products and individual surface elements are deformed;

    " At the beginning of surface damage the plastic deformation occurs - please take a look at https://doi.org/10.1051/itmconf/20171506003 and https://doi.org/10.1088/1757-899X/710/1/012016 in this papers this phenomenon for steel is well shown. Moreover, you have to justify the "oxide sheets" removal. It can be interesting to show it in the paper or support this phrase by literature. 
  5. I disagree with the "grains spalling" phrase. You wrote in further sections of your paper that "individual grains spall" - from a materials science point of view, it is impossible to remove the grains ... probably you meant the next to deformation, the process of material erosion (please, take it into account that it is difficult to extract individual grain from the ferrous matrix... please improve this statement. 
  6. Please explain the motivation for pressing fig 10 in the methodology section. Now, it is unclear. 
  7. I disagree with your statement: "
    This 463
    time corresponds to the end of the growth of erosional destruction.

    ", simply the cavitation erosion cannot be stopped while the cavitation loads continue - please improve this phrase... 

Minor comments 

  1. The first phase of introduction: explain what kind of "ultrasonic liquid treatment" you meant, provide some examples.
  2. This phrase should be clarified "
    a significant area of erosion damage, which practically does not change 70
    during further treatment

    " - what future treatment do you mean? 
  3. In the whole manuscript change "erosion caverns" to "erosion pits"
  4. Phrase "
    Such pressures cause plastic deformation and erosion of the treated surface.

    " - you should mention about material fatigue - cavitation erosion has fatigue nature
  5. L44 - missing "lambda" sign
  6. Instead word "cavities" use "pits" in the whole manuscript
  7. Table 1 - improve "40X".
  8. Clarify this phrase: "
    the roughness parameter Ra changed less than 256
    5% 3 measurements successively

    "
  9. "
    n the principle of feeling the rough of 267
    the measured surface

    " - unclear, simply, you probably meant "stick profilometer technique"
  10. What was the radius of the diamond needle tip used for roughness measurements?
  11. Unclear phrase "
    providing geometric 288
    and mechanical similarity of indentations as the indenter deepens under load.

    "
  12. Specify the hardness conditions. What was the load and dewelling time?
  13. Rather than "different part of specimen" write "cavitation threated areas"
  14. instead of "
    further treatment does not 324
    lead to a change in the structure.

    " write: "...in the surface roughness"
  15. Begining of the seciton 3.2 should be moved to "methodology".
  16. this phrase: "
    II - the 423
    beginning of the spalling of grains

    " must be improved. It is impossible to spall the grains - you meant the erosion of material or plastic deformation...
  17. Figs 5-10  should be referenced in the manuscript text. Now, they are not. It must be improved. 
  18. Please transfer this phrase to methodology "
    Figure 9 shows the dependence of the change in oil-capacity Q on time during 455
    cavitation treatment. This parameter is not a roughness parameter, but it is entirely 456
    determined by the surface microrelief, and it is a characteristic of the actual area of the 457
    researched surface. Therefore, oil-capacity is important in coating, for example, and it 458
    will determine an adhesion strength of coatings.

    "
  19. 308 x308 nm area of roughness measurement is too limited area to correctly state the roughness parameters on relatively rough surfaces 
  20. a relatively regular profile with uniformly 483
    alternating
     - this phrase is unclear - you polished your samples before testing
  21. This phrase should be improved : 
    the dynamics of changes in the microstructures of 503
    steels
    - you did not conduct the microstructural investigations....
  22. The 7th conclusion does not relate to the manuscript and should be deleted.

Author Response

Thank you very much for a detailed and constructive review of the paper. We carefully considered all your comments and made changes to the manuscript, and also tried to answer the questions that arose. We hope that the revised version of the paper will receive your approval.

Corrections for your comments in the text are highlighted in green

Reviewer 2 Report

This paper describes the study of the effect of ultrasonic cavitation on the surface of steels. Two different steels are studied and the changes observed at the surface during treatment are described (structure, roughness, hardness, etc.). In general the paper is interesting, the results are original and the experiments are well done. Nevertheless, I found a lot of typo and syntax erros that should be corrected before final publication, maybe the editor can help for this point. I also find the paper long and maybe some parts can be shortenned. I also have questions that would enhance the quality of the paper. I recommend publication after the following comments will be taken into account. 

Introduction:
- a comment: the first sentence sounds not good to me since it is not only the liquid treatment that generates the surface layer of metal products. Ultrasound propagates and may generate acoustic cavitation that can influence chemicals in solution and play a role at the surface (jets, streamings, shock wavec, etc.). Therefore, I recommend changing the first words of the sentence "Ultrasonic liquid treatment, based on [...]" by "Acoustic cavitation, based on [...]". And then, you can begin the second paragraph, with " Acoustic cavitation consists in the formation of bubbles [...]. 

- there was maybe a problem during the conversion from doc to pdf because the sign "gamma" does not appear within the text (p2, line 44). Also, I don't understand the values given in the brackets line 45 p2, maybe it is the value of the polytropic index ?

- even if it is not the purpose of your study, the conditions given in your introduction for P and T relate to the hot spot model dealing with the quasi-adiabatic heating of the gas inside the bubble. First, I find the related references 1-3 quite old. Then, recent studies demonstrated the formation of a non-equilibrium plasma inside the cavitation bubble. Please see for instance Nikitenko et al. Ultrason. Sonochem. 2017, 35, 623-630.

-Page2, line 77: the authors indicate "Currently, the process is well studided with a significant number of publications...". The refered publications are all from the 60's, please update the references or modify the text. For instance, Belova et al. Chem Science 2013; Belova et al. ACS appl. Mater. Sci. 2015 have observed and discussed interesting properties for related interfaces. In addition the introduction is very long, some parts could be shortened for clarity. Please comment.

-Also, I find that really recent publications dealing with the effect of acoustic cavitation on different materials could have been taken into account due to the link with the author's manuscript:
For instance: Work on magnesium: Ji et al. J. Hazardous Materials 2021; Ji et al. Ultrason Sonochem 2018. Work on titania: Zhukova Ultrason. Sonochem. 2017; Work on glass or silicon: Virot et al. J Phys Chem C 2012 and J Phys Chem C 2010. 

- The second part of the sentence p3 l129 looks strange to me, a verb is missing ? "In this regard, the relevance of studies 129 on the effect of ultrasonic liquid treatment on the surface properties at the micro- and 130 nanolevels has increased, and the effect of these properties on the features of the 131 operation of specific types of products." 

- same remark line 137 p3: "showed an increase in the roughness 137 parameter Rtm: when treatment of Al after 4 minutes - over 30 µm; when treatment of Cu 138 after 5 minutes - 17.9 µm; when treatment of Ag after 5 minutes - 13.3 µm."

-"this work, studies on the use of ultrasonic cavitation-erosion treatment for 188 widely used structural steels 40Kh (DIN, EN: 41 Cr 4; GB: 40 Cr; AISI: 5140) and 45 have 189 been carried out in order to study the possibilities of changing their surface propertie". Why are you only giving the composition of 40 kH, what about 45 ?

-l 199 p5 "one of them remained the control and the second was 199 subjected to ultrasonic treatment." Maybe better to say how you treated the sample. If you applied similar treatment as the one you applied with ultrasound, it is better to say you applied similar conditions without ultrasound. 

-l 206 p5, I have a problemen with the composition of the transducer which should reach 100%, you only have 49Co and 2V ? maybe 49 Fe ?

-The sample-sonotrode distance is indicated to be 4 mm within the text and 2 mm on your scheme. PLease correct. Also, how did you leasure this distance ? please indicate it. 

-Did you sonicated in air or under another saturating gas ? please indicate it and the rate if needed. Also did you measure the pH of the solution before/after experiment? 

-l 259p6: "treatmented specimens" I suppose you want to say "treated specimens" ?

-"without treat of specimens under vacuum", maybe better to write "without vaccum conditions"

-l 306 p 8 Why is the erosion occuring at grain boundary preferentially ? is it related to surface energy in comparison to one observed on grains ? please comment

-l 319 p 8 Why is the main mechanism for the growth of the area related to an increase of the size of the previously formed cavitites ? Could it be related to hetergeneous nucleation of acoustic bubbles ? please comment.

-Figure 4: HV should be defined. 

-What is the standard error for hardness measurement (to be added in the manuscript) ? (in other words, is the variation noticed between ca. 160 and 180 HV for 40kH sample significant ?

-l 371 p8: you are comparing 40X and 45 steels while 40 kH is used in the rest of manuscript. PLease homogeneize.

-I am not familiar with the different steels and I would have expected a description of the differences for both samples. Even if the composition is given, what are their behaviour ? why are they used ? for which application ? what is the effect of the small variation in chemical composition given in the table on their properties ? why did you choose them ? and finally what is the influence of these properties on your results and observations ?

-L 429-433 p 13: the following sentences are redundant: "The dynamics of changes in other measured roughness parameters for both steels is largely correlated with the Ra parameter. 
Thus, the roughness parameters Rz and Rtm (Figure 6) are similar to the changes in Ra." Maybe you could delete the second sentence and replace the first by: "The dynamics of changes in other measured roughness parameters (Rz and Rtm, Figure 6) for both steels is largely correlated with the Ra parameter."

-L 458 p 14, what is the oil-capacity parameter ? please define it and explain how you calculated it in the experimental section. 

-Figure 12 caption: "treatment" instead of "treatmet"

-Finally, why are you observing such different behaviour for your samples ?

Author Response

Thank you very much for a detailed and constructive review of the paper. We carefully considered all your comments and made changes to the manuscript, and also tried to answer the questions that arose. We hope that the revised version of the paper will receive your approval.

Corrections for your comments in the text are highlighted in yellow

Round 2

Reviewer 1 Report

Dear Authors, thank you for your improvements. I accept your explanations.

I have one remark about your future works. I think that you should use 2um or max 5um radius of diamond tip used for roughness measurements. It seriously improves the roughness results.

Author Response

Thank you very much for your suggestion. Now we are working on improving the instrument base and have already bought a sensor with a diamond tip with a radius of 2 microns, which we will use in future work.

Reviewer 2 Report

In general, I appreciate the modifications made by the authors. Nevertheless the authors only modified the text for some questions. I think that the manucript could be improved with few sentences and explanations (see my first review) about the mechanism of erosion under ultrasound (heterogeneous nucleation), the different behaviour between both surfaces, and the updated references. I recomment publication after those minor points wille be taken into accout. 

Author Response

Thank you for your appreciation on this article! In accordance with the comments, we have made changes to the manuscript. Provisions were added on the mechanism of erosion under ultrasound, the different behaviour between both surfaces, and references was also corrected (reference 1-3,7). Thanks for your comments!